# “*It Makes You Feel That You Are There*”: Exploring the Acceptability of Virtual Reality Nature Environments for People with Memory Loss

**DOI:** 10.3390/geriatrics6010027

**Published:** 2021-03-12

**Authors:** Noreen Orr, Nicola L. Yeo, Sarah G. Dean, Mathew P. White, Ruth Garside

**Affiliations:** 1European Centre for Environment and Human Health, University of Exeter Medical School, Knowledge Spa, Royal Cornwall Hospital, Truro, Cornwall TR1 3HD, UK; N.Orr@exeter.ac.uk (N.O.); nlyeo@outlook.com (N.L.Y.); 2Clinical Trials Unit, College of Medicine and Health, University of Exeter, Exeter EX1 1TE, UK; S.Dean@exeter.ac.uk; 3Cognitive Science Hub, University of Vienna, 1110 Vienna, Austria; mathew.white@univie.ac.at

**Keywords:** memory loss, virtual reality, technology, nature environments, qualitative research, dementia, long-term care

## Abstract

**Aim:** To report on the acceptability of virtual reality (VR) nature environments for people with memory loss at memory cafes, and explore the experiences and perceptions of carers and staff. **Methods:** A qualitative study was conducted between January and March 2019. Ten adults with memory loss, eight carers and six volunteer staff were recruited from two memory cafes, located in Cornwall, UK. There were 19 VR sessions which were audio recorded and all participants were interviewed at the end of the sessions. Framework analysis was used to identify patterns and themes in the data. **Results:** During the VR experience, participants were engaged to varying degrees, with engagement facilitated by the researcher, and in some cases, with the help of a carer. Participants responded positively to the nature scenes, finding them soothing and evoking memories. The VR experience was positive; many felt immersed in nature and saw it as an opportunity to ‘go somewhere’. However, it was not always positive and for a few, it could be ‘strange’. Participants reflected on their experience of the VR equipment, and volunteer staff and carers also shared their perceptions of VR for people with dementia in long-term care settings. **Conclusions:** The VR nature experience was an opportunity for people with memory loss to be immersed in nature and offered the potential to enhance their quality of life. Future work should build on lessons learned and continue to work with people with dementia in developing and implementing VR technology in long-term care settings.

## 1. Introduction 

Interaction with natural environments, especially marine and coastal, can benefit health and wellbeing [1]. However, many older adults, including those in long-term care, encounter barriers to accessing nature [2] which may contribute to widespread feelings of boredom and depression [3]. How residents might connect with nature to reduce these negative symptoms and improve wellbeing is receiving increased attention from researchers [4]. One possibility is to simulate aspects of nature indoors, enabling access for those residents who may only experience the outdoors infrequently or not at all [5]. Exposure to nature via large TV screens has produced some therapeutic benefits for residents with dementia including improvements in mood [6] and heart rate, a physiological indicator of stress [7]. While more recent work has found that fully immersive head-mounted Virtual Reality (VR) offers even greater benefits than TV [8], there has been a lack of research using totally immersive VR involving older adults with dementia in real world settings [9].

VR can be thought of “…as a way to relocate people to virtual places and take part in events and activity there” (p. 29, [10]. Participation is key to differentiating VR from other forms of human-computer interaction, in that the person “…*participates in the virtual world rather than uses it*” (p. 5, [10]). As Heim (p. 70, [11]) puts it, “…VR *insists* that we move about and physically interact with artificial worlds” (italics added). A special feature of VR is sensory immersion [11] which is the ‘technical goal’ of VR: to substitute ‘real’ sensory stimuli (e.g., visual, auditory, olfactory, haptic) with computer-generated ones (p. 4, [10]). However, VR may not achieve this in practice as typically, VR has been primarily an ‘optical technology’ (p. 1, [12]) with sound sometimes considered. Arguably, visual stimuli are the easiest type of sensory stimuli to replicate in VR and may be sufficient in inducing a sense of immersion by itself in some applications (p. 4, [10,12]), possibly because vision is often regarded as the dominant sense [13]. A combination of technologies such as head-mounted displays, headphones with sound/music, and hand-held controllers providing haptic feedback are used to deliver the immersive VR experience. Ultimately, the aim of a VR experience is ‘presence’, that feeling of ‘being there’ in the virtual environment, despite knowing that one is not actually there [14].

There are two primary types of VR: the first is the 360° video made of real scenes filmed with a series of special panoramic lenses facing in different directions that are then brought together to produce a 360° full surround experience; and the second is computer generated images (CGI), often created using gaming engine software. The 360° video represents a relatively passive form of VR with the user observing pre-recorded footage, and requiring little interaction. While greater interaction and immersion is possible with CGI, it is also computationally more demanding and (at least when the study was conducted) requires an accompanying laptop and a heavier, more cumbersome headset. In contrast, the 360° videos can be played on a standard mobile phone placed into a much simpler, lighter and less complicated headset. It was the latter type of VR that was selected for the VR nature experience intervention.

The long-term goal of this research was to develop a VR nature environment intervention for people with dementia living in long-term care facilities. At the study’s inception, we were not aware of any research with this population that had specifically trialled exposure to nature by immersive 360-degree videos shown via a head-mounted display. Since the use of VR in healthcare is a relatively recent development, its use with people with dementia is not well understood [9]. There are potentially a number of issues that could limit its use such as cybersickness, fear of, or unwillingness to use, new technology, and the ability to participate and experience presence. Therefore, in order to understand acceptability of VR technology with people with dementia, user testing is crucial (p. 1720, [15]). Before undertaking an extensive formal trial in long-term care settings, we piloted the VR nature experience with older people with mild to moderate memory impairment living in the community. Given that most existing studies on technology and people with dementia acknowledge that the technology is often used as a ‘joint activity’ in the ‘presence of another person’ (family member, carer or therapist) [16], we included carers and memory café volunteer staff in our study. In this first attempt to deliver the intervention, acceptability, defined as “…a subjective evaluation made by individuals who experience…an intervention” (p. 10, [17]), was assessed during and after delivery.

The aim of the study was to explore the acceptability of VR nature environments for people with memory loss at memory cafes, and explore the experiences and perceptions of carers and volunteer staff. The key research questions for this study were: How did older people with mild to moderate cognitive or memory impairment experience the VR nature environments intervention?What did carers and volunteer staff facilitating the memory café sessions think about VR for people with dementia and care home residents?

## 2. Materials and Methods

### 2.1. Study Design and Participants

Older adults with mild to moderate cognitive or memory impairment, their carers, and volunteer staff were recruited from two memory cafes in Cornwall, South West England. The memory cafes were chosen as a setting for the study because people that attend have memory issues and may or may not have a formal diagnosis of dementia. The researcher (NY) made four visits to each memory café, participated in sessions, and met with the memory café visitors and volunteers to explain the research project to potential participants. All participants provided informed written consent (and the University of Exeter Medical School Ethics Committee reviewed and approved the study (Jan19/B/192)).

### 2.2. Virtual Reality Nature Intervention

The 360° video VR nature intervention was designed to capture “restorative” nature elements such as gently breaking waves, dappled sunlight on water, thought to innately instil ‘soft fascination’, hold the viewer’s attention effortlessly, and thereby reduce negative emotions, promote relaxation and positive mood states [18]. However, pilot trials revealed that with a lack of ‘activity’ (e.g., people, animals, movement) in the 360° scenes, users quickly became confused and disengaged as to the purpose of the VR, as they expected ‘something to happen’. Given that one of the most problematic issues in long-term care is persistent lack of engagement among residents [19], any intervention should not risk causing further disengagement or confusion. In consultation with memory café chairpersons, we carefully chose clips containing identifiable landmarks, groups of people and interesting, non-threatening wildlife, against a backdrop of ‘tranquil’ scenery. 

The 360° video was a series of 5 × 30 second clips fading into one another, designed to feel like a day’s journey through a recognisable local spot, starting in the morning and ending with a sunset. The videos showed pre-recorded scenes of local Cornish beaches and coastal areas with audio of natural sounds such as gently breaking waves. In addition, one video depicted people engaged in activities on a beach with sounds of laughter and chatter. Another clip was taken underwater on a reef that was not local but was of high quality and included footage of gently shoaling fish. Example stills from the clips are shown in Figure 1. There was ‘surround sound’ rather than headphones to enable the carer to hear and converse with the person with memory loss where needed. 

The 360° video was played on ZTE Axon 7 smartphone, placed inside a Google Daydream^TM^ VR headset to allow fully panoramic viewing. The smartphone screen had a resolution of 1440 × 2560 pixels (Quad High Definition (QHD)). The smartphone was linked through a wireless connection to a Vodafone N8 Smart Tab 4G tablet using the Showtime VR app, from which the researcher could control the 360° video remotely. This allowed for a more seamless transmission, in which the researcher and carer (where present) could follow the 360° video content on the tablet screen in ‘real time’, enabling a shared experience. The Daydream headset (Figure 2) was fitted with wipe-clean face cushions which were cleaned with alcohol wipes between individual sessions, and laundered between café visits. 

### 2.3. Data Collection and Analysis

The video sessions were audio recorded in order to capture ‘in the moment’ reactions of the participants, sometimes difficult to reproduce in interviews. At the end of the sessions, semi-structured interviews (~10 minutes) were undertaken with the participants to explore perceptions of the of the VR nature experience, experiences of nature and technology, and how VR nature experiences might be suitable for people living with dementia and in long term care. Where both members of the dyad were interviewed together, the initial questions were directed to the person with memory loss, and then subsequently to the carer. 

The interviews were recorded and transcribed verbatim. The interview data were analysed using framework analysis [20], commonly used for the thematic analysis of semi-structured interview transcripts. It offers a flexible and systematic approach to analysing qualitative data [21]. Here, it enabled us to combine both deductive and inductive approaches to analysis, with some themes pre-selected based on the literature and the research questions, but open to the possibility of uncovering ‘unexpected’ themes from the data.

The transcripts were read and re-read, along with the interviewer’s reflective notes, as the first step in becoming familiar with the data. This helped to both identify recurring themes and concepts, and devise a conceptual framework of ‘main themes’. The next stage involved coding the transcripts and grouping the codes so that data with similar content were brought together. Then, a set of thematic matrices or charts were created which summarised and synthesised the data (Appendix A). It remained possible to add additional themes, if required, iteratively throughout the analysis process. Quotes from the participants provide ‘thick description’ allowing verification of these themes. The findings described below are supported with exemplar quotes and coded as volunteer (V), carer (C), or person with memory loss (ML), along with a numerical code, reflecting their participant number.

## 3. Results

There were 19 VR sessions at the two memory cafes: four sessions with people with memory loss; three with carers only; six with dyads (comprising an individual living with memory loss and their carer); and six with memory café volunteers (Table 1). Each participant tried the VR experience once. 


**Engagement during VR Experience**


The participants engaged with the VR nature environments to varying degrees, with many responding with quite detailed descriptions and commentaries on what they were seeing, and others commenting less and attempting to remove the headset before the end of the session. For example, one man with memory loss was animated throughout the experience:

I’m on the beach. Haven’t been on the beach in ages! Ooh, it’s lovely. Which beach is it, maid*?…Oh now I’m under the sea! Look at all these fishes. It’s good innit? Wow!…and now there’s someone comin’ in on a boat. Hello! Ahhh. The waves are lovely (10-ML) [*a Cornish dialect word for girl].

Another participant was similarly engaged and then after one and a half minutes stated that she had had enough: 

It’s sort of a forest thing, and it’s got a path…like mountains an’ that as well. It’s very nice…Oh wow I can see a beach now. Oh yeah, it’s a bit like Polzeath it is, with people on it. Oh yeah, people playing games an’ that on the beach. It’s hot ‘cause they got their shorts and t-shirts on. It’s very nice. Oh I can see a pink bucket as well. A…a very busy beach…very busy…oh yeah, very nice. [inaudible] they’re going out in the boat now. Oh yeah, there’s a chap going out on the beach—on the boat now. Ooh, very nice. I’ve had enough now (05-ML).

There were two others who had limited engagement as they removed the headset a number of times (for example, 07-ML removed the headset after approximately 40 seconds, and 08-ML removed the headset several times during the experience). Engagement was facilitated by the researcher (NY) and in some cases, with the help of a carer. Some carers chose to view the VR nature environments, or part of it, in advance of their partners, to encourage the person with memory loss to participate:

Oh that’s under the sea, oh that’s looking, oh that’s beautiful! Absolutely! Oh we like these sort of things. [husband’s name], you’ll enjoy this one. The coral, yeah, it’s lovely [husband’s name]. You’ll enjoy that (16-C).

She then continued to encourage her husband to describe what he was seeing and persuaded him to keep the headset on. 


**Reflections after the VR experience**


Participants reflected on their experiences in the interviews after the VR nature experience and six themes are expanded on below, with additional comments from volunteer staff and carers tabulated in Table 2. 

### 3.1. “It Makes You Feel That You Are There”: VR as an Immersive Experience 

The VR experience often centred around immersion (and presence), and the opportunity to interact with others in a virtual world.

Participants reported that they felt immersed in nature: ‘at the sea’, ‘on the sand’ and ‘feeling the sand’ (“…It looks real, and you feel part of it” [06-V], “…with that, virtually, you’re in it, aren’t you? It’s fantastic” [11-V]): 

Oh, now *I’m in the sea* with the fish!…Back on the beach. Who’s that over there? (13-ML)

It makes you feel as if you’re at the sea, right at the sea, doesn’t it?…it makes you feel that you *are there* [emphasis] on the sand, with the sea rolling in (15-V).

Another volunteer participant reflected on how the VR experience meant ‘you were there, in nature’:

and I’ve got a beautiful view where I live, so I can look out, but I’m looking through a window whereas that [the VR], you’re there, which is the difference, yeah (18-V).

One of the participants drew out that the feeling of presence made her ‘feel better’ as she had difficulty accessing beaches in her everyday life because of reduced mobility:

Some of these beaches I can’t access. So it’s actually quite nice to…see. This actually makes you, it does actually make you feel better ‘cause you can sort of, like *it feels like you can feel the sand* [laughs] (19-ML).

Another participant stated: 

I mean it makes you feel, good in a way, you know, just as you can see it, and um, *you don’t see anything else*, just all those fishes, the massive fish. (17-ML)

One participant observed that, despite the noisy environment of the memory cafe, she felt that she had ‘left’ the room, “…closing me ears off to that [the noise]. Concentrating on the panoramic views” (09-V).

A few of the participants found themselves responding to the situations they saw and described how they could have interacted with the people on the beach:

I liked seeing the people on the beach and I liked to, playing the Frisbee thing, you know the one on the beach and he was standing there and I thought, ‘well is he gonna talk to me?!’ you know [laughs] and I saw the chap in the water catching it and it was just nice to be ‘in’ there, especially when the surfer came down by your side and went down, it was just like being there (18-V).

### 3.2. “Oh Beautiful, It Was Lovely”: The Soothing Effect of Nature Scenes

All of the participants responded positively to the VR nature experience, typically describing it as ‘beautiful’ and ‘lovely’. It was clear that participants enjoyed nature and appreciated nature ‘views’ and ‘scenery’. Some added that they particularly enjoyed the colours in the scenes (“I’ve enjoyed looking at them. They’re lovely pictures and that. All the colours are beautiful” [01-C], “Oh, very pretty. I love the colours I do. Awww” [05-ML]).

A number of the participants highlighted aspects of the nature scenes that had particular appeal such as the sea, the beach, and the fishes:

The beach and the water coming in and the fish, was lovely [laughs] (08-ML).

Oh that sea, yeah. This is why we came back to Cornwall I think. Well it was, definitely. Incredible sand, isn’t it (02-C).

Oh the fishes were beautiful they were. I never been er…what d’ya call it?…Diving, I’ve never done it. But I love the water (10-ML).

Some described these different aspects relaxing and calming: 

the sea, the one in the sea was beautiful. Well I like water, for a start. It was just, um, the colours were very beautiful and erm, soothing and um…I like watching fish (12-ML).

### 3.3. “It Was Really Nice, Nice to See Places I Recognised”: Using Nature to Reminisce

Many of the participants appreciated the familiarity of the scenes and recognised local places in Cornwall: 

it was really nice, nice to see places I recognised. I mean it’s always nice, when you can identify things—you do it don’t you, even when you’re watching the telly and that sort of stuff (14-V).

The scenes also prompted many to talk about their past experiences of nature or past experiences in general. One participant said “…in a way it brings back things to you. You know, going to the seaside, the sea coming in. I think perhaps, that’s it, it brings back the memory of it” (06-V), and another stated:

Oh on the beach, you know, because I love the coast, I do. I grew up here in C…well not here, bit further down, place called Crantock, near Newquay… spent many a summer holiday in Crantock. My Mother was still there, see. And we took the kids, and then the grandkids when they come along… It’s [the coast] part of me, you know, growin’ up with it. I do miss it terribly (13-ML).

Memories could also be tinged with sadness in that the participants realised that there were places that they could no longer enjoy, illustrated by this excerpt from an exchange between a person with dementia and his carer: 

16-ML: I took her a lot of places, didn’t I maid*? [*a Cornish dialect word for girl]

16-C: Yeah, but I mean, it was interesting to see the different places, an’ that

16-ML: Yeah

16-C: Which we won’t be seein’ anymore, anyhow will we?

One participant with dementia was prompted by the scenes of the underwater coral reefs to recall when he had seen the corals in Egypt in the 1950s. He described how he had been in the British Army doing his National Service and had gone down the Gulf of Suez and swam in the sea where there were “lots of little fishes” and coral (16-ML).

### 3.4. “It’s…Like Goin on Holiday”: VR as a Different Experience 

For some participants, the VR experience offered new or different experiences and an opportunity to ‘go somewhere’:

but it was lovely ‘cause it gives you, I don’t know, a sense of freedom, yes. Anyone who’s had to be indoors, they just get that feeling of being there. And certainly when it was just the sea, the empty sea, I’d just like to go and have a nice swim (18-V).

the going under the sea. And just being able to look around like you’re walking somewhere—like you’re *going* somewhere. It looks, it’s really good, yeah (19-ML).

Similarly, seeing the fish evoked a sense of wonder at life underwater (“Oh now I got some fish. Beautiful isn’t it? There’s so many wonders underneath aren’t there, that we don’t know about…hidden treasures” [06-V]), and reminded one participant of the nature documentaries that she and her partner enjoyed: 

we like that coral one. Where you’re under the sea. Because we watch those programmes anyhow…and we love doing that, don’t we (16-C).

One participant described her experience as ‘going on a holiday’:

It’s very, very good…[inaudible] like goin’ on holiday, kinda thing, an’ it’s a bit like Polzeath, ‘cause, you got the sea, the sea there [gestures], you got the boat there [gestures] and the houses there, and the sand, and there’s people around you (05-ML).

Many participants commented on the people and animals, and appreciated the busy beach ‘activity’: 

Oh wow I can see a beach now. Oh yeah, it’s a bit like Polzeath it is, with people on it. Oh yeah, people playing games an’ that on the beach. It’s hot ‘cause they got their shorts and t-shirts on. It’s very nice. Oh I can see a pink bucket as well. A…a very busy beach…very busy…oh yeah, very nice. [inaudible] they’re going out in the boat now. Oh yeah, there’s a chap going out on the beach—on the boat now. Ooh, very nice (05-ML).

Another participant declared that she felt inspired to visit the places depicted in the VR nature experience (“Oh, inspired! [laughs]. I want to go and visit the places” [11-C]). 

### 3.5. Strange and Scary: The VR Nature Experience Is Not Always Positive 

There were some negative responses to the VR nature experience, finding them potentially confusing or strange, such as this carer: 

Well from what I saw of the little bits, when she [her Mother] was at the beach, she obviously quite liked that. I think under the water was a bit confusing for her.” (07-C) 

One participant said “Yeah, it’s quite strange actually, ‘cause it’s not *quite* real. It’s got a strange light to it. Sort of, edges on the figures and things” 12-ML), and at one point found it frightening:

12-ML: Back to the beach. Ooh, clouds. Oh the little group in the corner there… Ooh yeah. 

It’s a lot better down here than when we were there and I was up the top, terrified… Ooh, I don’t like this.

Researcher: What is it?

12-ML: It’s all steep and falling away into the sea below me. I’m sitting on a rock.

Researcher: Are you OK, do you want to take it off?

12-ML: No.

### 3.6. Responses to VR Equipment

Some participants found the VR technology amazing, describing it as ‘incredible’, ‘brilliant’, ‘clever’ and ‘remarkable’.

Now it’s gone right behind me. My neck won’t swivel that far. How on earth do they do this? Oh gosh it makes it look so good that I want to go in the sea. Extraordinary marks in the sand. Absolutely amazing [inaudible] (02-C). 

Oh wow yeah. It’s quite an amazing sort of thing. It’s almost like an experience without the smells [laughs] (04-V).

However, many of the participants discussed issues with the equipment which included their confidence in using the equipment, its heaviness, difficulties with body movement, and with focus while wearing glasses.

Only a few of the participants claimed that they experienced no discomfort with the VR equipment. Some participants with dementia were uncomfortable with the headset; one removed it after 40 seconds (07-ML) and another commented when fitting the headset: “It’s a bit heavy” (05-ML). This was echoed by a number of other participants and some said they would prefer something less ‘intrusive’ and more ‘user-friendly’ (07-C), that “actually sat on your head properly” (19-ML). Most agreed that they could only tolerate the headset for a short time (“…I think if you were to watch it for, if you watched a film on it for half an hour, or an hour, it would become a bit heavy then” (09-V). One participant suggested it could make the wearer feel nauseous: 

I could see that it could make you feel a bit queasy if you stayed in it for a long time. I think if it, it’s made smaller and more comfortable, then it would be beneficial (12-ML).

A few of the participants commented that they had problems with body movement and could not turn round to see everything:

the only problem was I wanted to be able to go round 180 degrees and I couldn’t. I think a wheelie chair would be a terrific advantage because I think a lot of people aren’t as mobile as I am, you know (02-C). 

That looked round a bit too hard for my neck (19-ML).

Some participants who wore glasses had problems with focus: (“perhaps I should have had it with me glasses on because it would have been a bit more focussed [11-V]); and one felt that her experience was spoiled: 

I found the first bit on the beach was quite blurred. I don’t…was that my eyes or was it?… I was having quite a job focussing on a lot of that… It just kind of spoilt, spoilt it for me a bit, because I couldn’t see some of the detail in anything (14-V).

One of the participants with dementia described a strange light around the “edges on the figures and things” and that people had looked “not quite real” and were a “bit too brightly coloured” (12-ML).

A small number of the participants living with dementia noted that they would not be able to operate the VR technology without help. One participant wondered if the researcher would always be available to help: 

I mean you put it on and took it off. Will you always be there to put it on and take it off? (03-ML)

In contrast, a few felt confident that they would be able to use the VR equipment (“It’s just pressing a couple of buttons” [19-ML]).

None of the participants living with dementia claimed to be using much technology in their daily lives (“Too old in the tooth for that!” [laughs] [08-ML]). One indicated that he had the expertise—a telecoms degree—but as his carer explained:

You used to use all these things but you don’t get along with it any more. I’m the one with the computer now. And mobile phone. You don’t telephone out. You barely turn on the telly unless I do it and find the channel [both laugh] (03-C).

For another couple without a computer, the carer explained, “We’re not interested. It isn’t our era if you know what I mean” (16-C). Other carers described technology—such as smart phones, tablets and computers—used in their everyday lives:

I’ve got my phone and my ipad, that sort of thing. And I get ‘round it alright. But I’m not expert in it. But you know, if I want to find out something, I usually can do it some way or another. Or phone up me brother in Australia and ask him [laughs] (11-C).

Café volunteers believed that they could use the VR technology (“I’m not high-tech like you youngsters but I’m not illiterate either [laughs], so yes, I think I could do it” [14-V]) but were unsure if older carers, or people with dementia, would be able to use it:

Maybe, maybe they would need someone with them all the time, if they didn’t, to set it up for them. ‘Cause I’m just thinking of how my husband was, and he wouldn’t have been able to do that. Because unfortunately, I mean, he would always put music on for us for Saturday night while we had a nice meal. And then one night he said, well, he said ‘I can’t get the CD to play’ and I said ‘well you just press that button’. You see, it had just gone out of his mind. He, well, he lost his sort of, spatial awareness, thing. (18-V)

### 3.7. Volunteer Staff and Carers’ Perceptions of VR for People with Dementia in Long Term Care

Generally, volunteer staff and carers felt that the VR nature experience was a ‘good idea’ for people with dementia in long term care settings. The perceived benefits included, accessing nature in an alternative way, relieving boredom, triggering memories, and improving mood and calming. However, they also had reservations, fearing that the person with dementia could find the VR experience potentially confusing and unable to cope with the equipment. Table 2 provides a summary of volunteer staff and carers’ perceptions of VR for people with dementia in long term care.

## 4. Discussion

The purpose of this study was to explore the acceptability of VR nature environments for people with memory loss at memory cafes, and volunteer staff and carers’ perceptions of VR for people with dementia in long-term care. The findings identified six themes around experiences and perceptions of the VR nature environment intervention. The VR experience was positive for participants, with a reported sensation of ‘presence’, the feeling of ‘being there’ on the beach or underwater as depicted in the VR scenes. That feeling of ‘being there’ whilst knowing that ‘you are not actually there’ has been referred to as ‘place illusion’ and it is this illusion that creates the ‘wow’ factor (p. 38, [10]). For some participants, the feeling of being transported to, for example, a local beach, evoked a sense of wonder and amazement. There is no sense of presence in physical reality [10], a notion that was captured well by the participant who contrasted the VR nature experience with the beautiful nature views from her home. Where participants responded to the events and people on the beaches as if they were real, they experienced the illusion that Slater and Sanchez-Vives (p. 5, [10]) refer to as ‘plausibility’. Participants described how they felt that individuals on the beach were looking at them and might even talk with them, provoking their desire to interact, and indicating that they found the environment ‘sufficiently credible’. These findings suggest that, for some of the participants, the VR nature environments intervention delivered an experience that generated an “illusory sense of place and an illusory sense of reality” (p. 5, [10]). 

These findings support those from existing studies which found that people with dementia are able to perceive a sense of presence [22]. However, the sense of presence was not positive for all. With participants responding realistically to VR, it is hardly surprising that one participant reacted to her experience of ‘sitting on a rock’ with fear. This clearly affected her enjoyment and it is crucial to avoid participants feeling fear and anxiety, filming scenes that are sensitive to participants’ needs. Whether the people with memory loss perceived a sense of presence was likely affected by the extent of their engagement with the VR experience, and the degree of their cognitive impairment; the more advanced their impairment, the greater the possibility they may have found it ‘confusing’ [6]. 

Participants reported not only the perception of ‘being there’ but also the feeling of ‘being away’, a key factor in Attention Restoration Theory [18] which posits that nature distracts from the everyday and mundane. ‘Being away’ had a dual perspective; one was experiencing familiar places that were no longer easily accessed, and the other was experiencing new places such as ‘going on holiday’ or ‘going somewhere’. The familiarity of the VR nature scenes gave enjoyment to participants as they sought to identify exact locations and to relate stories and memories, often linked to particular places. By interacting with each other—the person with memory loss and the carer—participants were often able to create meaning around the ‘virtual’ places and connect them to their own personal lives. Arguably, this ‘place-based reminiscence’ helped participants to “re-experience past places” (p. 5, [23]). Such reminiscence could invoke joyful memories such as seaside holidays with children, but there could be a poignancy too, as participants reflected on how places were no longer accessible to them, usually because of transport and mobility issues. The potential of the VR nature experience to stimulate reminiscence could help people with dementia maintain continuity of the self [23,24] particularly important for people with dementia living in long term care who may feel this is broken. 

VR nature also offered participants the opportunity to experience new places, or nature in a ‘new’ or different way. Participants found the scenes inviting them—to swim in the sea and to come away on a holiday, and perhaps, if neither of these were realistic options, then the virtual experience was an acceptable alternative. The scenes with the underwater coral reefs elicited a feeling of ‘going somewhere’, somewhere very different for the participant who had never been a good swimmer but who would have liked to explore ‘under the sea’ and never had the opportunity. Thus the VR nature experience gave such participants “…the possibility to step outside the normal bounds of reality and realise goals in a totally new and unexpected way” (p. 2, [10]).

The beauty of the nature scenes, with views of beaches and the sea, was highlighted by all, and was key to their engagement and enjoyment. Whilst immersed and contemplating the beauty of nature, some participants noted a calming and soothing effect, suggesting that virtual nature can induce a restorative experience. Brown, Mitchell [25] argue that a sense of wellbeing—which can reduce stress—should be the main ambition for meaningful activity in advanced dementia, and call for creative and innovative interventions that can improve quality of life for people with advanced dementia. Arguably, the VR nature environments shows promise for providing a ‘stimulating’ yet ‘familiar environment’ that engages people with dementia in a ‘unique way’ (p. 2, [9]).

## 5. Limitations

One limitation in this study was that the environment in the memory cafes was ‘not ideal’; the VR nature experience was set up in a room where other activities were taking place and background noise could be a distraction for participants, as was the case with one participant who observed she would have liked to have heard the sounds of nature such as the ‘roar of the sea’. Anderson, Mayer [26] noted the importance of reducing distraction from background noise when using VR but arguably, it could be challenging to be free of background noise in an aged-care environment.

Another potential limitation is that the VR world is a fast moving one and equipment and types of simulation are rapidly changing and developing; Sayma, Tuijt [9] found that differences in software and hardware availability could ‘dramatically alter’ the experience of participants in studies that were just one year apart. Since completion of this study, light-weighted headsets have been released and used successfully with participants with mild cognitive impairment [27]. Recent, and no doubt future, developments will make CGI immersion more straightforward for use with various populations. Financial constraints also impacted decisions on technology; for example, participants were unable to interact with the VR nature environments as we could not readily combine photorealism and interactivity within the budget constraints of the study.

## 6. Implications

The findings suggest that the VR natural environments intervention did engage a number of the participants. The researcher facilitated engagement, often with the help of a carer, so it is likely that for people living in long-term care, staff facilitation would be a requirement [6,22]. Similarly, it is unlikely that people with more advanced dementia would be able to use the technology without support from staff and carers [23]. The need for one-to-one support would be time-consuming for care home staff, causing some in the study to doubt if it is practical to introduce this technology into care homes for individuals with more severe dementia [16]. Future work should focus on the feasibility of implementing the VR nature experience in long-term care settings and should include assessing time consumption for one-to-one interaction and training requirements for care home staff. In future research, it would also be beneficial to understand the extent of study participants’ cognitive impairment via standardised tests.

Perhaps the key issues for the technology of the VR nature experience were practical; many participants experienced discomfort with the headset, and as already noted, this is an area where technology improvements are ongoing [27]. However, issues with the HMD may mean that the VR nature environments intervention is ‘not for everyone’. A few participants reported visual issues and, as people with dementia may have visuo-perception problems, this should be taken into account in future development. The experience was also limited for a few participants by being unable to turn through 360 degrees in their chair which could be addressed by using a 360° swivel armchair.

## 7. Conclusions

This study shows that people living with mild to moderate cognitive impairment who participated in a VR nature intervention had a sense of presence in the nature environments. Despite the importance of presence in VR being recognised [28], few studies on immersive VR in dementia have attempted to establish participant presence [9]. Given that presence is a subjective experience, the qualitative approach used in this study contributes to a developing field of research where evidence is still limited. It also enhances our understanding of the wellbeing benefits that people with mild to moderate memory loss can derive from a VR nature intervention, and shows promise for those living in long-term care. Since people with advanced dementia in residential care are less involved in activities [29,30], the VR nature intervention could be a way of engaging them in meaningful activity—with support from carers. Future work should build on lessons learned from this study and continue to work with people with dementia in developing and implementing VR technology in long-term care settings.

## Figures and Tables

**Figure 1 geriatrics-06-00027-f001:**
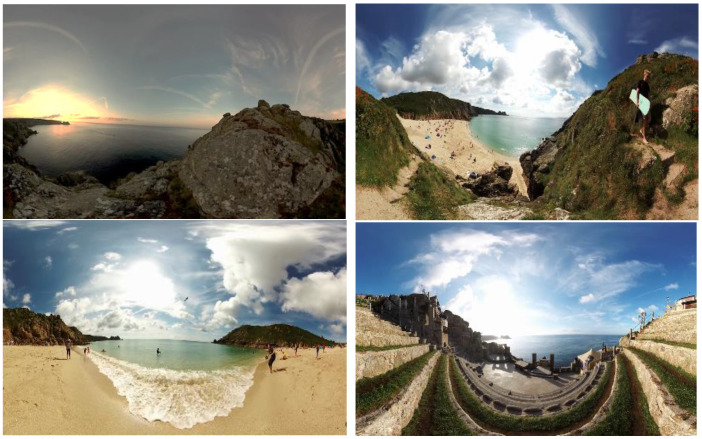
Example stills from each of the 30-second 360° video clips.

**Figure 2 geriatrics-06-00027-f002:**
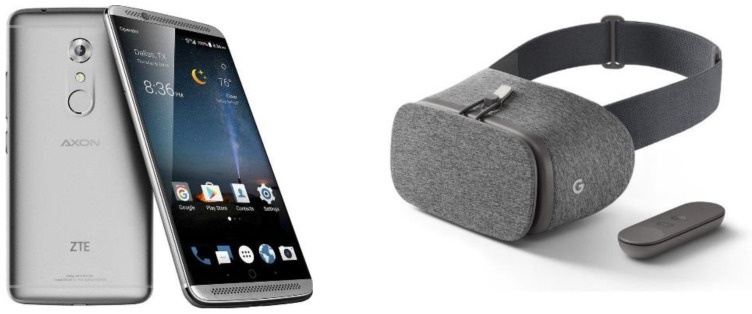
Daydream Headset and ZTE AXON 7 Smartphone.

**Table 1 geriatrics-06-00027-t001:** Profile of Participants.

	People with Memory Loss (ML; *n* = 10)	Carers (C; *n* = 8)	Volunteer Staff (V; *n* = 6)
**Gender**			
Women	6	7	5
Men	4	1	1
**Relationship to person with memory loss**			
Spouse		5	
Daughter		2	
Friend		1	

**Table 2 geriatrics-06-00027-t002:** Volunteer staff and carers’ perceptions of VR for Person with Dementia in long-term care.

Perceptions of VR	Illustrative Quote from Volunteer (V) or Carer (C)
1. Alternative means of accessing nature and the outdoors ^+^	“I think it’s a great idea, especially, like you say, for people who can’t get out.” (17-C) “I think maybe for the person that’s being cared for, then yeah. Sort of make them feel like they’ve got out of the four walls … even if it’s ten minutes.” (07-C) “sometime in the future, I’ll not be able to drive, and I think, if I couldn’t get out, that would be great, that sort of thing… ‘Cause it brings the outside back right into you, yeah.” (09-V)
2. Relieving boredom ^+^	“It gives them a chance to go and see something…she gets quite bored, and so I think going out to look at something like that, would be quite nice. Whether she’d sit there for long with it on…but you don’t have to do you” (17-C)
3. Trigger memories ^+^	“Oh, I think that could be marvellous. If you can make it personal, …trigger the memories, you know, it would be lovely.” (10-C) “I think it depends on what sort of dementia they’ve got, really…I think it’d be lovely for a lot of people. Especially, as you say, the ones that are in homes. You know, especially if it was a horrible afternoon and you could say ‘alright we’ll go down to the beach, this is what it was like’. And bringing back memories.” (18-V) “…if they’re able to watch it, I think it’d be brilliant for them. Just like, the way I say, to be able to bring back memories for them. Especially if it was all, sort of, local to that person’s… well, you know, we live in Cornwall, so they’re places in Cornwall that we would recognise.” (14-V)
4. Improve mood and calming ^+^	“I suppose it could help put people in a better mood, especially if they’re havin’ a bad day, and they can’t get out.” (10-C) “I do think that if someone’s having a really bad day, that it would help to calm them. It’s like music, ‘cause seeing that visually—the sun and people having fun on the beach…” (06-V)
5. Potentially confusing ^−^	“I don’t think it would help him [husband LwD]. He’d go through it. He wouldn’t remember what had happened two minutes afterwards. I think he might be a bit confused.” (02-C) “I mean for some people they would find it a bit confusing, you know, and think: ‘what on earth’s going on here…this is a bit weird’. Um and especially those who, you know, their memory loss is so great that they don’t remember one minute to the next minute…that contradict themselves all the time you know, so you ask them if they enjoy it one minute and they’d go ‘yes’, and then you ask them again and they’d go ‘what?’… they can’t even remember what they look…saw” (04-V)
6. Unable to cope with the VR equipment ^−^	“I think one thing about the technology is definitely that [husband] wouldn’t be able to cope with it. He would definitely need somebody there to switch it on, switch it off, do whatever is necessary.” (03-C) “…it’s quite heavy on your face, and I think…it’s all going to vary depending on what sort of dementia it is, but I wonder how they would cope moving around, physically and that sort of thing, to see, or whether they would just sort of think they’ve just gotta look forward. And even, even if they realise they can move around, would they physically be able to do it easily, I don’t know?” (14-V)
7. Time-consuming for use in long term care ^−^	“I think the sad thing is though…whether it would actually get used…in a care setting, I don’t know. Because it’s time, it’s gonna be time consuming isn’t it, and it’s one-to-one, and everything, and maybe in care settings that’s quite a difficult task isn’t it, time-wise.” (14-V)

^+^ Positive; ^−^ Negative.

## Data Availability

The data presented in this study are available in the Appendix A.

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
