# Peer review of "It Makes You Feel That You Are There”: Exploring the Acceptability of Virtual Reality Nature Environments for People with Memory Loss"

_geriatrics, 2021, doi:10.3390/geriatrics6010027_

Round 1

Reviewer 1 Report

The video sessions are well described, but I am a bit concerned about the analysis. Did you use both deductive and inductive analysis simultaneously or did you first one and then the other? I do not have access to Ritchie’s book about framework analysis. In order to better understand the analysis, It should be valuable to know which pre-selected themes you used, and when in the analysis process you introduced them. In the beginning or at the end?

I’m also concerned about the way the findings are described. The large proportion of quotes in the descriptions under the themes contributes to this. A more conceptual abstraction would facilitate reading.

I would like to suggest that the staff’s and volunteers’ view on virtual reality should be separated in the result presentation. Otherwise, there is a risk that the most important result, the experience of people with memory loss, disappears.

Author Response

Reviewer 1 Comments

Response

The video sessions are well described, but I am a bit concerned about the analysis. Did you use both deductive and inductive analysis simultaneously or did you first one and then the other? I do not have access to Ritchie’s book about framework analysis. In order to better understand the analysis, It should be valuable to know which pre-selected themes you used, and when in the analysis process you introduced them. In the beginning or at the end?

The Framework Method for the analysis of qualitative data was developed by the National Centre for Social Research (NatCen) in the UK and has been used since the 1980s.

The thematic framework was constructed using deductive and inductive analysis, i.e. it included themes from the research questions and those that were identified in the data. The thematic framework was constructed after the initial ‘familiarisation’ stage of reading and re-reading the transcripts to gain an overview of the topics. Identifying and developing themes is part of the early stages of the analytic process when using the framework approach. We have added clarification on p. 6.

I’m also concerned about the way the findings are described. The large proportion of quotes in the descriptions under the themes contributes to this. A more conceptual abstraction would facilitate reading.

The quotes from the participants are the qualitative data providing the ‘thick description’ which enable the reader to verify for him/herself the conclusions reached by the authors. This is typical for a framework approach which is not so concerned with developing theory and a ‘conceptual abstraction’.

I would like to suggest that the staff’s and volunteers’ view on virtual reality should be separated in the result presentation. Otherwise, there is a risk that the most important result, the experience of people with memory loss, disappears.

Thank you for this suggestion. Including the carer and volunteer staff views alongside the people with memory loss was a decision made by the authors at the study’s conception.  

In the presentation of the results, we were careful not to obscure the experiences of people with memory loss and prioritised their experiences and perspectives (24 quotes) over those of the carers and volunteer staff (12 and 15 quotes respectively). Given that existing studies show that ‘another person’, whether a family member or carer, is generally required for the full facilitation of a technology intervention (Goodall et al, 2020), we believe that understanding their experiences alongside that of the person with memory loss is incredibly valuable (see p. 5).

Reviewer 2 Report

This paper reports on the acceptability of virtual reality (VR) nature environments people with memory loss at memory cafes, and explore the experiences and perceptions of carers and staff. To this end, they conducted an experiment with 12 participants affected with different degrees of memory lost over 19 sessions. 

Overall, the paper is well-written and presents some useful results for the acceptance of VR in their intended target group. 

There are a few elements that the authors can make it clearer:

  • What was the resolution of the phone? Resolution can affect people's perception of how realistic the virtual environments are. 
  • Why the authors did not choose to use a device that can provide sound? This could have improved the experience given that one of their limitations was that the study was conducted in a noisy environment. 
  • Why the authors did not choose virtual environments that are tailored to their participants? This could be useful to understand how these environments can help them bring back memories (that were perhaps lost).
  • Why the authors did not chose to allow the participants some degree of interactivity with the virtual environment? While participants could perhaps interact with them in a limited manner, it would be useful to see if and how they would and could interact with them.

One other aspect that the authors could consider is to add more information from the perspective of the the carers. Right now they are presented as supplementary material. It could be useful to have some more results in the body of the paper, given that in the abstract the paper also emphasizes on the perception of the carers.

Author Response

Reviewer 2 Comments

Response

What was the resolution of the phone? Resolution can affect people's perception of how realistic the virtual environments are.

The screen had a resolution of 1440x2560 pixels (Quad High Definition (QHD)). This is equivalent to other premium smartphones which were on the market at the time the research was done. This detail has been added on p. 6.

Why the authors did not choose to use a device that can provide sound? This could have improved the experience given that one of their limitations was that the study was conducted in a noisy environment.

The device did provide sound as stated on p. 5:  “The videos showed pre-recorded scenes of local Cornish beaches and coastal areas with audio of natural sounds such as gently breaking waves. In addition, one video depicted people engaged in activities on a beach with sounds of laughter and chatter.” That one participant could not hear the sound as it was drowned out by the background noise was mentioned in the Limitations section of the paper. We had ‘surround sound’ rather than headphones as we wanted the carer to be able to converse with the person with memory loss where needed, so they needed to be able to hear them. This additional detail can be found on p.  6.

Why the authors did not choose virtual environments that are tailored to their participants? This could be useful to understand how these environments can help them bring back memories (that were perhaps lost).

Tailoring virtual environments to each participant may well have evoked memories but the design cost precluded this as a viable option and it is unlikely to be within reach of settings like memory cafes or care homes for people with dementia. However, designing the virtual nature environments to include scenes of local beaches etc. did indeed enable participants to reminiscence (see p. 9).

Why the authors did not chose to allow the participants some degree of interactivity with the virtual environment? While participants could perhaps interact with them in a limited manner, it would be useful to see if and how they would and could interact with them.

We agree that this is a very good idea and future research could do this. At the time of this study, we could not readily get a combination of photorealism and interactivity within the budget constraints of the study. This is acknowledged on p. 14.

One other aspect that the authors could consider is to add more information from the perspective of the carers. Right now they are presented as supplementary material. It could be useful to have some more results in the body of the paper, given that in the abstract the paper also emphasizes on the perception of the carers.

Thank you for this suggestion. We have carefully considered the presentation of the Results in light of all the Reviewer’ comments (which to some extent contradict each other) and in particular, the suggestion that we reduce the Results section.  We agree that the perception of the carers and volunteer staff are important and feel that, while we have prioritised the views and experiences of the people with memory loss (24 quotes), we have also included a good representation of the carers’ and volunteers’ perspectives (12 and 15 quotes respectively).  

Reviewer 3 Report

The paper addresses the acceptance of 360°-videos presented via HMD to older adults with memory complaints, their caregivers, and volunteers.

The paper is generally well-written and deals with an interesting topic. Additionally, the use of a think-aloud protocol is emerging as a good way to capture participants’ opinions during and after the experience. However, I have some concerns that must be addressed prior to publication.

MAJOR ISSUES.

1 – Page 2, Lines 66-75. The authors said they preferred the 360°-video than CGI because of a simpler setup and a more light-weighted device. I do not agree with this assertation. First, there are many stand-alone devices on the market (e.g., Oculus Quest and HTC Cosmos) that do not require any additional laptop. These are all light-weighted and well accepted by different populations of users, including older adults with cognitive impairments (see Park JH, Liao Y, Kim DR, Song S, Lim JH, Park H, Lee Y, Park KW (2020) Feasibility and tolerability of a culture-based virtual reality (VR) training program in patients with mild cognitive impairment: A randomized controlled pilot study. Int J Environ Res Public Health 17, 1–9.). Please revise this concept.

2- At the end of the Introduction, authors should clarify what is the aim of proposing videos to people with memory impairment. Is it supposed to have any clinical or psychological benefit?

3 – I am not sure whether the authors aimed at evaluating the acceptance (of the technology, Venkatesh, V., & Bala, H. (2008). Technology acceptance model 3 and a research agenda on interventions. Decision sciences, 39(2), 273-315) or the acceptability of the intervention (i.e., the degree to which non-professional stakeholders found an intervention to be fair, reasonable, intrusive and consistent with treatment expectations (Wolf, 1978; Kazdin, 1980)). Please clarify this point.

4 – results obtained from care-givers and volunteers are interesting in terms of defining the acceptance of such an intervention, but they must not be mixed with MLs’ opinions. I believe that participants with memory loss must be the focus of the manuscript (as also indicated in the title), therefore I would suggest revising all Results section dividing clearly the results, and focusing more on patients’ opinions than on healthy people’s.  

5 – I believe that the discussed results cannot be interpreted …as positive. In a small sample of ML individuals, almost 1/3 of participants had issue with the HMD and the experience in general (one quitted after 1 minute and a half, and two removed the HMD). Thus, I would be more cautious in interpreting the study outcomes. Also, at Page5, lines 195, it is said that “engagement was facilitated …”; this seems to imply that was not easy to convince ML participants to try the HMD.

MINOR ISSUE

6 – please revise the beginning of Section 3. Results. What do the authors mean with sessions? If I understood correctly, each participant tried the HMD experience once. If so, this would go in the direction of evaluating acceptance (see Comment 3).

7 – It would be useful to know the extent of participants’ impairment, perhaps using a standardized cognitive test.

8 – I would also suggest to shorten the results’ presentation.

9 – You may add a reference for thematic analysis (e.g., Clarke, V., Braun, V., & Hayfield, N. (2015). Thematic analysis. Qualitative psychology: A practical guide to research methods, 222-248.)

Author Response

Reviewer 3 Comments

Response

Page 2, Lines 66-75. The authors said they preferred the 360°-video than CGI because of a simpler setup and a more light-weighted device. I do not agree with this assertion. First, there are many stand-alone devices on the market (e.g., Oculus Quest and HTC Cosmos) that do not require any additional laptop. These are all light-weighted and well accepted by different populations of users, including older adults with cognitive impairments (see Park JH, Liao Y, Kim DR, Song S, Lim JH, Park H, Lee Y, Park KW (2020) Feasibility and tolerability of a culture-based virtual reality (VR) training program in patients with mild cognitive impairment: A randomized controlled pilot study. Int J Environ Res Public Health 17, 1–9.). Please revise this concept.

Thank you for this reference which has been published since we completed the study. We appreciate that VR is a fast-moving field but the technology used was appropriate in terms of accessibility and cost when the study was conceived and carried out. The headsets mentioned by Park et al (2020) were released after we completed the study. We have noted this in the Limitations on p. 14.

At the end of the Introduction, authors should clarify what is the aim of proposing videos to people with memory impairment. Is it supposed to have any clinical or psychological benefit?

The aim was to trial an immersive virtual nature-based intervention with individuals living with memory loss. As we note in the paper’s introductory paragraph, exposure to nature has been shown to be associated with health/wellbeing benefits for this population, e.g. (White et al., 2018), including virtually, e.g. as in (Reynolds, Rodiek, Lininger, & McCulley, 2018), where nature videos were shown on a large TV screen. However, prior to this study, we were not aware of any research with this population that had specifically trialled exposure to nature by immersive 360-degree videos shown via a head-mounted display, and so the purpose of this study was to test the acceptability of this format of delivery (see p. 4-5).

I am not sure whether the authors aimed at evaluating the acceptance (of the technology, Venkatesh, V., & Bala, H. (2008). Technology acceptance model 3 and a research agenda on interventions. Decision sciences, 39(2), 273-315) or the acceptability of the intervention (i.e., the degree to which non-professional stakeholders found an intervention to be fair, reasonable, intrusive and consistent with treatment expectations (Wolf, 1978; Kazdin, 1980)). Please clarify this point.

Thank you for this observation. The study aimed to evaluate the acceptability of the intervention, both content and delivery. There was the possibility that the content and the VR kit could lead to confusion and discomfort for people with memory loss, both of which is acknowledged in the manuscript. By way of clarification, we have added a sentence on acceptability on p. 5 which follows Sekhon et al’s (2017) definition of acceptability for healthcare interventions, as the ‘subjective evaluation’ made by individuals who experience the intervention. 

results obtained from care-givers and volunteers are interesting in terms of defining the acceptance of such an intervention, but they must not be mixed with MLs’ opinions. I believe that participants with memory loss must be the focus of the manuscript (as also indicated in the title), therefore I would suggest revising all Results section dividing clearly the results, and focusing more on patients’ opinions than on healthy people’s. 

Thank you for this suggestion. Including the carer and volunteer staff views alongside the people with memory loss was a decision made by the authors at the study’s conception. Given that existing studies show that ‘another person’, whether a family member or carer, is generally required for the full facilitation of a technology intervention, we believe that understanding their experiences alongside that of the person with memory loss is incredibly valuable. That said, in the presentation of the results we were careful to ensure that the experiences of people with memory loss were the main focus (24 quotes), rather than those of the carers and volunteers staff (12 and 15 quotes respectively).

I believe that the discussed results cannot be interpreted …as positive. In a small sample of ML individuals, almost 1/3 of participants had issue with the HMD and the experience in general (one quitted after 1 minute and a half, and two removed the HMD). Thus, I would be more cautious in interpreting the study outcomes. Also, at Page5, lines 195, it is said that “engagement was facilitated …”; this seems to imply that was not easy to convince ML participants to try the HMD.

Thank you for this comment. The HMD was an important aspect of the acceptability of the intervention and we received important feedback. It is clear that the issues with the HMD may mean that the VR nature environments intervention is ‘not for everyone’. We believe that we have presented a balanced interpretation of the results and clearly state in the Abstract, for example, that the VR nature environments intervention was not always a positive experience. 

All of the participants with memory loss consented to participate. As explained in the methods section, the researcher visited the memory cafes as part of the lead-up to the data collection and met with volunteer staff at the memory cafes and with people experiencing memory loss and their carers. It is important to note that memory cafes exist for both the people living with memory loss and their carers and that they attend together, so conversations about the research took place with both.

The importance of ‘facilitating’ the engagement of people with memory loss/dementia with technology has been acknowledged in a recent systematic review on the use of technology for people living with dementia (Goodall et al, 2020).

– please revise the beginning of Section 3. Results. What do the authors mean with sessions? If I understood correctly, each participant tried the HMD experience once. If so, this would go in the direction of evaluating acceptance (see Comment 3).

Each participant tried the virtual nature experience once. At the beginning of the Results section, we reported the total number of VR sessions carried out at the two memory cafes.

– It would be useful to know the extent of participants’ impairment, perhaps using a standardized cognitive test.

We agree that this could be useful and future research could certainly do this, noted on p. 14. It was not part of this study design, as approved by the University of Exeter Medical School Ethics Committee. We used a convenience sample of participants in the community as a first step to piloting the VR nature environments experience and did not want to add to the ‘burden’ of participating in the research project. Furthermore, memory cafes are set up in direct opposition to the clinical model of dementia care in that anyone who identifies as having memory loss can attend. Therefore, in line with the memory café ethos, we did not consider it appropriate to introduce a clinical assessment as part of this work.

I would also suggest to shorten the results’ presentation.

We have carefully considered the Results section in light of all of the Reviewers’ comments. Feedback that the Results should include more information (particularly on carer and volunteer staff perspectives), has led us to conclude that we should not reduce the Results section.

You may add a reference for thematic analysis (e.g., Clarke, V., Braun, V., & Hayfield, N. (2015). Thematic analysis. Qualitative psychology: A practical guide to research methods, 222-248.)

We did not reference thematic analysis (and in particular, Braun & Clarke’s approach to thematic analysis) as we used the framework approach to analyse the qualitative data. The analytic approach is similar to thematic analysis but it would be incorrect to insert the suggested reference.

Round 2

Reviewer 1 Report

I have no further comments

Author Response

Thanks for your comments.

Reviewer 3 Report

1) "While greater interaction and immersion is possible with CGI, 71
it is also computationally more demanding and requires an accompanying laptop and a heavier, more cumbersome headset. In contrast, the 3600 videos can be played on a standard mobile phone placed into a much simpler, lighter and less complicated headset." - I understand the point of view of the authors, and that they acknowledge a limitation of their study, but this sentence is not correct, and must be modified.

2) I still believe that the presentation of Results (and consenquent Discussion) is not reflecting well the reality. If, on the one hand, I am aware of the importance of care-givers' and volunteers' opinions, on the other, the focus must be on the persons needing such an intervention. Presenting their comments mixed together does not allow to draw appropriate conclusions (i.e., the fact that 1/3 of participants with ML had issue with immersive VR).

As mentioned by the authors, this lack of acceptability is not necessarily an issue, but it just means that VR may not be for everybody. This fact must be discussed in this terms. 

- I would also suggest specifying that in each session, each participant tried the immersive experience once.

Author Response

Reviewer 3

Response

"While greater interaction and immersion is possible with CGI, 71 it is also computationally more demanding and requires an accompanying laptop and a heavier, more cumbersome headset. In contrast, the 3600 videos can be played on a standard mobile phone placed into a much simpler, lighter and less complicated headset." - I understand the point of view of the authors, and that they acknowledge a limitation of their study, but this sentence is not correct, and must be modified.

We have modified the sentence as follows:

“While greater interaction and immersion is possible with CGI, it is also computationally more demanding and (at least when the study was conducted), requires an accompanying laptop and a heavier, more cumbersome headset.” (see p.2)

We have also added the following sentence to the Limitations (p.12): “Recent, and no doubt future, developments will make CGI immersion more straightforward for use with various populations."

I still believe that the presentation of Results (and consenquent Discussion) is not reflecting well the reality. If, on the one hand, I am aware of the importance of care-givers' and volunteers' opinions, on the other, the focus must be on the persons needing such an intervention. Presenting their comments mixed together does not allow to draw appropriate conclusions (i.e., the fact that 1/3 of participants with ML had issue with immersive VR).

In addition to our previous response, we emphasise that the participants with memory loss were the focus of our study and we were careful to prioritise their perceptions and experiences in the presentation of the findings (24 excerpts from people with memory loss, 12 from carers and 15 from volunteer staff). The six themes were developed through analysis of transcripts from people with memory loss in the framework analysis. Excerpts from carers and volunteers which also fitted these themes were added as appropriate, but none of the themes in the main text were based solely on their views.

For the authors, carer and volunteer staff experiences and perceptions were considered as very important from the inception of the study.  We noted this on p.2 indicating that existing studies on technology and people with dementia have shown that technology is often a ‘joint activity’ and in the presence of others such as carers and family members. This also fits with a broad view of dementia care (Smebye & Kirkevold, 2013) that embraces relationships, interconnectedness, and working in partnerships with family and professional carers. It is worth noting that other review feedback indicated that there could have been greater focus on the carer and staff perspective and we do not feel that including their perspective prevents the reader drawing appropriate conclusions. We reported the issues that people with memory loss encountered with the VR very clearly in the Results section, and comments from carers and volunteer staff could reinforce these points. The issues were also considered in the Limitations and Implications sections of the paper.

As mentioned by the authors, this lack of acceptability is not necessarily an issue, but it just means that VR may not be for everybody. This fact must be discussed in this terms.

We agree that the VR nature environment intervention is not for everyone and note that despite technological improvements in HMD, it may be that some will never be comfortable with HMD (p.12).

- I would also suggest specifying that in each session, each participant tried the immersive experience once."

Thank you. We have specified that each participant tried the VR experience once (p.5).